# Molecular Breeding for Abiotic Stress Tolerance in Crops: Recent Developments and Future Prospectives

**DOI:** 10.3390/ijms26189164

**Published:** 2025-09-19

**Authors:** Mario A. Pagnotta

**Affiliations:** Department of Agricultural and Forest Sciences, Tuscia University, Via S. C. de Lellis, snc, 01100 Viterbo, Italy; pagnotta@unitus.it

**Keywords:** stress, tolerance, molecular breeding, genome editing, hormonal regulation, genome-wide association, machine learning, Nature-Based Solutions (NBS), nanotechnology

## Abstract

The document is an updated review, starting from the Special Issue “Molecular Breeding for Abiotic Stress Tolerance in Crops” published in the Int. J. Mol. Sci. It reviews molecular breeding strategies to enhance abiotic stress tolerance in crops, addressing challenges like drought, salinity, temperature extremes, and waterlogging, which threaten global food security. Climate change intensifies these stresses, making it critical to develop resilient crop varieties. Plants adapt to stress through mechanisms such as hormonal regulation (e.g., ABA, ethylene), antioxidant defense (e.g., SOD, CAT), osmotic adjustment (e.g., proline accumulation), and gene expression regulation via transcription factors like MYB and WRKY. Advanced tools, such as CRISPR/Cas9 genome editing, enable precise modifications of stress-related genes, improving tolerance without compromising yield. Examples include rice (*OsRR22*, *OsDST*) and wheat (*TaERF3*, *TaHKT1;5*). Epigenetic regulation, including DNA methylation and histone modifications, also plays a role in stress adaptation. Specific studies focused on polyamine seed priming for improved germination and stress resistance, cadmium detoxification mechanisms, and genome-wide association studies (GWAS) to identify genetic markers for salt tolerance and yield. Research on salinity tolerance in wheat emphasizes sodium exclusion and tissue tolerance mechanisms. Future perspectives focus on genetic engineering, molecular markers, epigenetic studies, and functional validation to address environmental stress challenges, including the use of AI and machine learning to manage the large amount of data. The review underscores the importance of translating molecular findings into practical applications to ensure sustainable crop production under changing climates.

## 1. Plants and Abiotic Stresses

Abiotic stresses, encompassing drought, salinity, temperature extremes, and waterlogging, pose a significant threat to global food security, disrupting plant morphological, biochemical, and physiological processes and leading to diminished crop productivity worldwide [1]. Plants, being immobile organisms, must withstand a variety of environmental challenges, which frequently impair their distribution, impede their growth and development, and reduce agricultural output [2]. Drought stands out as a primary cause of crop yield loss, especially in regions that rely on rainfall, where unpredictable precipitation patterns heighten the risk of crop failure, posing a significant challenge for small-holder farmers who strive to maintain sufficient crop production. Climate change exacerbates the prevalence and intensity of abiotic stresses, thereby jeopardizing global crop production and productivity. In addition, climate change brings extreme, erratic events that alternate periods of drought with excess water and consequent waterlogging. Another important issue considered in the Special Issue “Molecular Breeding for Abiotic Stress Tolerance in Crops” is the increase in salinity, often because of drought, and the consequent increased use of artesian wells, which decreases water levels in the soil, especially for irrigated crops such as horticultural ones. But also, the increase in the salt wedge, that is, the sea entrance in the waterways.

Abiotic stress triggers significant changes at the physiologic, proteomic, metabolomic, and genomic levels in plants [3]. Plants have evolved complex mechanisms to perceive stress signals and trigger adaptive responses at the molecular level [4]. These mechanisms involve a composite interplay of signaling pathways, gene regulation, and metabolic adjustments that enable plants to cope with adverse environmental conditions. The study to gain knowledge and understand these mechanisms took advantage of using advanced molecular tools. In addition, molecular tools have revolutionized plant breeding by enabling breeders to dissect the genetic basis of stress tolerance and accelerate the development of climate-resilient varieties offering a promising strategy to improve abiotic stress tolerance in crops [5].

There are several possible mechanisms for tolerating stress. These include key hormonal regulation, antioxidant defense, osmotic adjustment, and how plants could regulate the expression of the involved genes. Efficient recovery after stress, such as rehydration, is essential for survival under temporary stress conditions [4] (Figure 1).

### 1.1. Hormonal Regulation

The role of phytohormones, particularly abscisic acid, is a significant signaling molecule in drought stress responses [6]. This insight is paving the way for a deeper understanding of how these hormonal pathways function under different stress scenarios (Figure 2). **Abscisic acid (ABA)** plays a central role in stress tolerance, particularly under drought and salinity conditions [7,8,9]. Genes like *SlPYL2*, *SlABI3*, and *SlABI5* are upregulated to enhance stress tolerance, while ABA degradation genes such as *SlCYP707-A2* are regulated to maintain ABA levels. ABA acts by closing Stomata to reduce water loss. In Arabidopsis, ABA signaling genes (*PP2C*, *SnRK2*) were downregulated, potentially influenced by master transcription factors like AHL20, PBF, and MNB1A [10]. **Ethylene (ET)**, a gaseous phytohormone, modulates growth and senescence, playing a pivotal and multifaceted role in mediating plant responses to a diverse range of abiotic and biotic stresses such as drought, extreme temperatures, and heavy metal toxicity, significantly influencing plant growth, development, and survival mechanisms [11]. Specifically, ethylene’s signaling pathway, involves endoplasmic reticulum-bound receptors, i.e., CTR1 kinase and the EIN2 protein, is crucial for transducing stress signals into appropriate cellular responses [12]. **Salicylic acid (SA)** plays an important regulatory role in multiple physiological processes, it induces systemic acquired resistance and triggers the expression of pathogenesis-related proteins Several genes functioning in SA biosynthesis, conjugation, accumulation, signaling, and crosstalk with other hormones [13]. In the *Arabidopsis* genome the *ICS1*, *ICS2*, *PAL1*, *PAL2*, *PAL3*, and *PAL4* genes have indicated that participate in the synthesis of SA [13]. **Methyl Jasmonate (MeJA)** is a volatile organic compound derived from **jasmonic acid (JA)** and plays a critical role in plant defense mechanisms against several stresses [14], modulating stress responses and activating defense-related gene expression. Exogenous methyl jasmonate reduces Na^+^ buildup and modulates K^+^ allocation between roots and shoots, thereby lowering the shoot Na^+^/K^+^ ratio. Increased K^+^ in shoots under salt stress sustains photosynthesis, metabolism, and osmotic balance while mitigating ion toxicity [15]. Beyond its role in stress alleviation, it also influences various physiological and developmental processes, such as growth, senescence, and reproduction [16]. **Auxins** mediate plant adaptations by coordinating differential signal transduction pathways and influencing key developmental changes in response to both internal and external stimuli [17]. For instance, they influence the synthesis of osmolytes and antioxidant enzymes, crucial components in mitigating cellular damage under drought conditions [18]. Their roles extend from modulating cell expansion and organ patterning to influencing overall root and shoot architecture, fundamentally impacting a plant’s ability to cope with adverse conditions [19]. **Cytokinins** are integral to numerous physiological processes, orchestrating a complex network of signaling pathways that enable plants to adapt and survive under adverse conditions. They play central roles in cell division, shoot development, nutrient allocation, delay of senescence, and regulation of stress responses. Their action is highly context-dependent and strongly shaped by crosstalk with other hormones such as auxins, ABA, ethylene, SA, and JA. **Gibberellins (GAs)** promote growth; they are often down-regulated under stress to conserve energy [5,20]. In stress response, they interact with other hormones (notably ABA) to fine-tune plant responses to environmental stresses. ABA and GA are antagonists during seed dormancy and germination, while auxin synergistically enhances GA biosynthesis and signaling [20]. See Section 3.1 for other Gibberellins’ roles and descriptions.

The hormons interact each other with **crosstalk** mechanisms to reach the best regulation and defense for plants. The synergistic interplay between Jasmonic Acid (JA) and ethylene (ET), abscisic acid (ABA), salicylic acid (SA), and gibberellin (GA) to regulate plant growth, abiotic stress tolerance, and defense resistance [10,21]. **ABA signaling can modulate JA** signaling pathways. For example, the transcription factor PBF interacts with JA signaling genes like MYC2, COI1, and JAZ [22]. **SA and JA** pathways often exhibit antagonistic interactions, SA signaling may suppress JA signaling under stress conditions [13]. **SA and ET** pathways can interact antagonistically or synergistically depending on the stress context. In Hordeum vulgare, β-Aminobutyric Acid (BABA) treatment downregulated SA biosynthetic genes (ICS2, EDS5, CM) and ET biosynthetic genes (ACS), while upregulating ERF2, which may act as a negative regulator of SA signaling [10]. **ET and JA** pathways often work synergistically in defense responses. In Hordeum vulgare, BABA treatment upregulated JA biosynthetic genes (LOX, TIFY9) and signaling genes (COI1, JAZ), along with ERF2, which positively regulates JA-responsive defense genes [10]. **ABA and ET** signaling can interact in stress responses. In both species, BABA treatment upregulated EBF1/2, negative regulators of ET signaling, suggesting a suppression of ET responses [10].

The hormone crosstalk is also different in the different plant stages. For example, Virag et colleagues [23], using high-throughput RNA sequencing (RNA-Seq), analyzed the transcriptomic changes across four distinct developmental stages of soybean flowers grown under field conditions. Hormonal signaling pathways were dynamically modulated across the stages. Auxin and gibberellin were prominent in early stages, promoting organogenesis and floral initiation [24]. The Cytokinin activity peaked during organ differentiation, while ABA, ethylene, and JA became dominant during later stages, particularly at anthesis and senescence. The Virag et colleagues [23], study provides a high-resolution temporal atlas of gene expression, offering insights into how hormonal crosstalk and environmental signals shape reproductive transitions. This knowledge could inform breeding strategies to optimize flowering time, improve yield, and enhance stress resilience in soybean and other crops.

### 1.2. Antioxidant Defence

Most of the stress in plants can induce a surge in the production of reactive oxygen species, thereby perturbing cellular equilibrium. Reactive oxygen species (ROS), including superoxide radicals, hydrogen peroxide, and hydroxyl radicals, are intrinsically reactive. ROS play a dual role in plants: although excessive accumulation under stress can cause oxidative damage, they also function as key signaling molecules, relaying stress signals to the nucleus through redox-regulated mitogen-activated protein kinase (MAPK) cascades. When oxidants are present in excessive concentrations, they can inflict substantial damage on essential biomolecules such as proteins, lipids, and nucleic acids, culminating in cellular dysfunction and ultimately, cell death [25,26,27]. To counteract this, plants have evolved complex metabolic pathways to detoxify and process these harmful compounds, thereby maintaining cellular homeostasis and ensuring robust growth and development [28]. This defense system encompass enzymatic and non-enzymatic components that work in concert to neutralize ROS [29] and mitigate their deleterious effects [30].

The **enzymatic antioxidants** components include a suite of highly efficient enzymes such as superoxide dismutase (SOD), catalase (CAT), peroxidase (POD), reductases, and ascorbate peroxidase (APX); also, protective enzymes such as glutathione S-transferase, ferritin, polyphenol oxidase, and alternative oxidase play crucial roles during stress. These enzymes can work independently or synergistically to maintain redox homeostasis and mitigate oxidative damage caused by reactive oxygen species (ROS) [31,32] and maintain cellular redox homeostasis [33]. The NADPH oxidase, also known as the respiratory burst oxidase homolog, situated on the plasma membrane, and the opening of voltage-gated calcium channels are crucial signaling inputs in plants [34]. Enzymatic antioxidants represent a critical line of defense in plants, orchestrating the detoxification of reactive oxygen species through a suite of specialized enzymes, including superoxide dismutase, catalase, and various peroxidase isoforms [33]. Several studies reported the enhanced activity of antioxidant enzymes, such as SOD, CAT, APX, and glutathione peroxidase (GPX) [35,36,37,38].

The **non-enzymatic** arsenal consists of small molecules such as ascorbic acid (AsA), glutathione (GSH), alkaloids, phenolic acids, α-tocopherol, carotenoids, amino acids, flavonoids, anthocyanins, vitamin E, and mineral elements, all contributing to the neutralization of oxidative threats [26,31]. The Ascorbate-glutathione pathway actively participates in the detoxification of hydrogen peroxide (H_2_O_2_, a type of ROS) and keeps the ROS under control [29]. The precise AsA regulates important cellular functions and developmental processes, including cell division and cell wall development. It improves the plant nutrient uptake, growth (both leaf and root), photosynthesis, and decreases lipid peroxidation. Furthermore, the precise regulation of these antioxidant systems is critical, as reactive oxygen species also function as vital signaling molecules involved in various physiological processes, necessitating a delicate balance between their production and detoxification [39]. This dual role underscores the sophisticated nature of plant redox biology, where controlled generation of reactive oxygen species can elicit beneficial adaptive responses to stress, whereas unchecked accumulation becomes detrimental [40].

### 1.3. Osmotic Adjustment

Plant osmotic adjustment is a critical adaptive mechanism for survival under various abiotic stresses. It is essential to comprehend how soil water potential is influenced by water deficit and salinity stresses. As established, hydric stresses lead to osmotic, ionic, and oxidative stresses.

Recent findings indicate that plants possess advanced osmotic stress perception and signal transduction pathways that regulate the expression of stress-responsive genes (Figure 1). Soil water potential decreases due to water deficit and salinity stresses, which leads to osmotic, ionic, and oxidative stresses [41]. Plants possess sophisticated osmotic stress perception and signal transduction pathways that govern the expression of stress-responsive genes [42]. Osmotic adjustment is vital for maintaining cellular turgor and facilitating water uptake, primarily involves the accumulation of osmolytes, which are broadly categorized into organic solutes and inorganic ions [43]. So, osmotic adjustment stands as a pivotal physiological mechanism in plants, enabling them to withstand and adapt to environmental stresses, particularly those related to water availability, such as drought and salinity, effectively lowering the osmotic potential and facilitating water uptake from the surrounding environment [44,45]. The synthesis and accumulation of osmolytes, such as proline, glycine betaine (GB), amino acids, sugars (e.g., T6P), alcohols, and quaternary ammonium [46,47,48], helps maintain cellular osmotic balance, turgor pressure, and protects cellular structures under stress, providing the necessary gradient for water absorption, and sustain cell turgor through osmotic adjustment [49]. This adjustment is essential for cell expansion, stomatal regulation, and overall plant growth [50] allowing plants to preserve turgor and facilitate water uptake. With the result of plants’ adaptation to osmotic stress and drought [47,51]. Additionally, the maintenance of chlorophyll levels and photosynthetic efficiency has underscored their importance in enhancing stress resilience [20,36,52,53]. This adaptive process also involves the synthesis of protective proteins and the activation of antioxidant systems to counteract oxidative stress [54]. For instance, trehalose reduces oxidative stress by improving antioxidant enzyme activity [48]. It is also possible to use an exogenous application of osmolyte, such as proline or sugars (Trehalose), to alleviates the adverse effects of stress on the growth and productivity of crop plants [55,56].

### 1.4. Gene Expression Regulation

Stress-responsive genes are upregulated or downregulated to adapt to environmental challenges [4,5,20,36,57,58,59,60,61]. The control and regulation of gene expression are tightly regulated by exposure to stress. This control is achieved by different molecular mechanisms that are highly dependent on the particular stress and organism [62]. Examples include genes involved in ABA signaling, chlorophyll biosynthesis, antioxidant enzyme production, and late embryogenesis abundant (LEA) proteins [36,63]. Transcription factors such as SlMYB78-like interact with other proteins (e.g., SlDREB3) to regulate stress responses via signaling pathways [4] (see also Section 3.5). The enhanced activity of ROS-scavenging systems reduces oxidative damage, as evidenced by lower levels of malondialdehyde (MDA) and hydrogen peroxide (H_2_O_2_) [4,36,64].

Moreover, mechanisms such as sodium exclusion prevent excessive sodium accumulation in leaves, while tissue tolerance allows plants to accumulate sodium without suffering damage [31,65,66]. Stress signals are transmitted via membrane-localized receptors, Ca^2+^ influx, and activation of protein kinases and transcription factors, leading to the expression of defense genes [5,57].

**Epigenetic Regulation** by the mean DNA methylation and histone modifications alters gene expression, contributing to stress adaptation and transgenerational stress resistance [36,57,67]. It was also studied that the possibility of stress imprinting through seed priming creates a “priming memory”, enabling plants to respond more effectively to subsequent stress [36,68].

## 2. Advanced Tools to Improve Tolerance

### 2.1. Genome Editing

Conventional breeding has significantly increased crop yields but has achieved limited success in improving tolerance to abiotic stresses [69,70]. This limitation arises partly from breeders’ tendency to evaluate genetic materials under optimal conditions. Moreover, the complexity of abiotic stresses and the varying sensitivity of plants across developmental stages make it difficult to establish reliable selection criteria for stress tolerance. Consequently, alternative strategies are needed to enhance crop resilience, yield, and quality. One such approach is genetic engineering [70], which enables the development of transgenic crops better equipped to withstand rapidly changing environmental conditions.

Genetic engineering and genome editing tools have emerged as promising tools for engineering abiotic stress tolerance in crops [71]; since they offer powerful avenues for developing crop varieties with enhanced resilience to various environmental stress [72]. These biotechnological approaches facilitate the targeted manipulation of plant genomes, enabling the introduction of novel traits or the precise modification of existing genetic information to improve stress tolerance [73]. This involves an array of techniques, including the overexpression of target genes, downregulation using RNA interference, virus-induced gene silencing, and advanced genome-editing tools like CRISPR-Cas9 [74]. CRISPR/Cas9 becomes a powerful tool enhancing precision in genetic modification. This is difficult to achieve through conventional breeding, enables significant alterations in crops with a fast genome editing to improve abiotic stress tolerance in plants [75].

By precisely targeting and modifying specific genes, genome editing enables breeders to fine-tune plant traits and enhance adaptation to changing environmental conditions. Genome editing using the CRISPR/Cas9 knock-out system has produced several crops with improved environmental stress tolerance. Some examples of the CRISPR/Cas9 use in different plant species, aiming to improve tolerance to specific stresses, are listed in Table 1.

### 2.2. AI and Machine Learning

Machine learning is a group of computerized modeling approaches that can learn patterns from data to make automatic decisions without programming explicit rules. It is a branch of artificial intelligence (AI), which has emerged as a transformative technology, permeating various aspects of modern life from personalized recommendations to self-driving cars. However, its potential within the realm of plant biology, particularly in understanding and mitigating abiotic stresses, remains relatively underexplored [108], even if it acquires increasing relevance, and particularly in plant breeding, is rapidly expanding [109,110]. In fact, the integration of AI-driven methodologies, encompassing machine learning and deep learning techniques, offers unprecedented opportunities for analyzing complex plant stress mechanisms and for predicting stress responses [111].

The complex interplay between plants and their environment, particularly under adverse conditions caused by abiotic stresses, has profound implications for plant physiology, agricultural productivity, and ultimately, global food security, with a pressing need for comprehensive insights into the complex molecular mechanisms governing plant stress responses to facilitate the development of climate-resilient crops [112]. The integration of machine learning methodologies into plant phenotyping pipelines presents an unprecedented opportunity to accelerate the identification of stress-responsive genes, decipher intricate signaling networks, and predict plant performance under diverse environmental scenarios [113]. By leveraging machine learning algorithms, researchers can analyze vast datasets encompassing genomic, transcriptomic, proteomic, and metabolomic information, coupled with physiological and phenotypic measurements, to uncover hidden patterns, predict complex interactions, and ultimately, engineer crops with enhanced resilience to abiotic stresses [76,114,115] thereby facilitating the identification of stress-resistant genotypes and the optimization of agricultural practices [113,116]. The application of computer vision and image processing techniques has revolutionized plant phenotyping, enabling non-destructive assessment of plant traits with unprecedented precision and throughput [117]. Convolutional Neural Networks and Recurrent Neural Networks have become indispensable tools for analyzing plant images, facilitating the identification of disease symptoms, quantification of growth parameters, and assessment of stress-induced morphological changes [118]. The development of automated plant disease detection techniques, incorporating image processing approaches at various stages, highlights the potential of machine learning to address critical challenges in agriculture [119,120]. The inherent complexity of plant stress responses, governed by intricate regulatory networks and influenced by a multitude of environmental factors, poses a significant challenge to traditional reductionist approaches.

The integration of vast datasets and discerning complex patterns is revolutionizing agricultural practices by enabling advancements in areas such as early disease detection, precise yield forecasting, and efficient resource allocation [121]. For example, in the field of precision farming, machine learning creates an interconnected network of devices that collect data with the help of sensors, drones, GPS monitors, and exchange the collected information to elaborate decision and activate automation. Machine learning technologies were used to predict the level of aflatoxin in corn by Branstad-Spates et al. [122] and an accuracy of 90.32% was observed. One of the machine learning tools is ANN (Artificial Neural Networks) that could be successfully used for studying the effects of atmospheric pressure, precipitation, temperature, crop disease, and Multilayer Perceptron-ANN (MLP-ANN) [123]. Imaging sensors can detect the onset of harmful effects before visible symptoms appear. Among available techniques, hyperspectral imaging is particularly valuable because it captures detailed plant reflectance data across a broad light spectrum (well beyond human vision) and records far more than the three-color bands used in conventional digital imaging. This capacity makes it a preferred method for detecting and classifying early stages of abiotic stresses and foliar diseases in plants, both in laboratory and field settings [124]. By capturing images with cameras, AI and machine learning technologies enable the monitoring of key soil characteristics, such as quality, fertility, microbial activity, and nutrient deficiencies, as well as plant growth patterns. Through advanced image analysis, AI can process and interpret this information far more rapidly than humans, making it possible to assess crop health, predict yields with greater accuracy, and detect signs of malnutrition [125].

This technological synergy is crucial for developing robust crop cultivars that can withstand the challenges posed by changing climatic conditions, ensuring global food security [110] (Please see also Section 4.1).

### 2.3. Nature-Based Solutions

Nature-Based solutions (NBS) approach is gaining considerable traction as an eco-friendly alternative to conventional chemical treatments, offering a sustainable pathway to enhance crop resilience against diverse environmental stressors [126]. Biostimulants, broadly defined by as “a product stimulating plant nutrition processes independently of the product’s nutrient content with the sole aim of improving one or more of the following characteristics of the plant or the plant rhizosphere: (a) nutrient use efficiency; (b) tolerance to abiotic stress; (c) quality traits; (d) availability of confined nutrients in soil or rhizosphere”; they could be substances or microorganisms, and represent a promising avenue for sustainable agriculture [127,128]. These biological formulations, which include a wide array of compounds ranging from simple organic molecules to complex living microorganisms, operate through various mechanisms such as bolstering antioxidant defenses, modulating hormonal pathways, and inducing metabolic adjustments within the plant system [129]. Unlike synthetic chemicals, whose indiscriminate use poses significant risks to ecosystem health and human well-being, biostimulants offer a safer and more environmentally benign method for improving plant vigor and stress tolerance [130]. The diverse origins of biostimulants, encompassing extracts from bacteria, fungi, seaweeds, amino acids, and humic substances, which are applied via seed coating, foliar sprays, fertigation, or soil amendments, necessitate a comprehensive understanding of their biological underpinnings for the development of scientifically sound applications and regulatory frameworks [131,132]. This rapidly evolving field seeks to leverage natural biological processes to mitigate the detrimental effects of climate change and other anthropogenic pressures on agricultural productivity. Specifically, microbial inoculants and plant-derived extracts exemplify the potential of biostimulants to enhance plant performance by optimizing nutrient utilization and bolstering intrinsic stress-response mechanisms [133]. This includes their established role in mitigating the adverse impacts of drought, salinity, and extreme temperatures, thereby contributing to food security in vulnerable regions [134] (Table 2).

### 2.4. Nano-Thecnologies

The exogenous application of molecules to induce stress tolerance could be better developed by nano-formulations [141]. Nanotechnology, particularly using nanoparticles, presents a promising approach for sustainable agriculture by enhancing crop adaptation mechanisms and overall stress tolerance [142].

Nano-formulations, composed of nanoparticles typically sized between 1 and 100 nm, provide notable benefits for improving the delivery and effectiveness of bioactive compounds in biological systems [143]. Their nanoscale dimensions enable them to cross plant surface barriers such as the leaf cuticle, stomata, and cell walls, thereby ensuring efficient transport into plant tissues [144]. In addition, their unique physicochemical properties, including high surface-area-to-volume ratio, tunable porosity, and controlled release kinetics, enable enhanced uptake efficiency and targeted action within the plant system, thereby minimizing environmental dispersion and optimizing resource utilization [145]. Nano-trehalose, for instance, may be applied either as a seed coating or as a foliar spray, exploiting its capacity to pass through stomata and promote cellular water retention other examples are listed in Table 3.

These advanced materials interact with plants at the cellular level, influencing key physiological processes and modulating defense responses to enhance stress resistance [146]. The combination of osmolyte strategies and nanotechnology is seen as an innovative method for improving crop resilience and maintaining agricultural productivity amid climate challenges. Enhancing trehalose, chitosan-proline, or other molecules delivery systems through nanotechnology and additional genetic improvements offers considerable promise for real-world agricultural applications [48,141]. These properties have led to the emergence of nanobiotechnology as a critical field, applying nanoscale tools and materials to address agricultural challenges and improve crop performance under adverse conditions [147]. It involves the intelligent design of nanomaterials, such as nanoparticles, nanofibres, nanoemulsions, and nanocapsules, which serve as sophisticated platforms for the precise and controlled delivery of agrochemicals and vital macromolecules directly to plant tissues [148].
ijms-26-09164-t003_Table 3Table 3Some Examples of nanobiotechnology applications in plants.ApplicationNanomaterial(s)Main EffectsExample CropsRef.NanofertilizersNano-hydroxyapatite, ZnO, urea nanoparticlesControlled nutrient release, higher uptake efficiency, reduced leachingMaize, wheat, rice[149]NanopesticidesSilver nanoparticles (AgNPs), CuO NPs, polymeric nano-carriersTargeted delivery of active ingredients, suppression of pathogens, reduced toxicityTomato, rice, wheat[150,151]Stress protectantsZnO, TiO_2_, SiO_2_ nanoparticlesEnhanced tolerance to drought, salinity, heavy metals via antioxidant defense and ion homeostasisWheat, rice, soybean[152]Nanocarriers for gene deliveryCarbon nanotubes, mesoporous silica nanoparticlesDNA/RNA/CRISPR delivery, improved transformation efficiency, transient expressionSpinach, arugula, tobacco[153]NanobiosensorsGold nanoparticles, quantum dots, carbon nanotubesEarly detection of pathogens, nutrient deficiencies, and stress markersVarious (e.g., virus detection in crops)[154]Postharvest nanocoatingsChitosan nanoparticles, AgNP-chitosan filmsProlonged shelf life, reduced microbial spoilage, maintained fruit qualityStrawberry, tomato, banana[155]


## 3. SI: Molecular Breeding for Abiotic Stress Tolerance in Crops

The topic of the Special Issue (SI) “Molecular Breeding for Abiotic Stress Tolerance in Crops” is a very complex one, not only because of stress complexity but also because of the several mechanisms that plants adapt to tolerate the stresses. This Special Issue, with its three reviews and six research articles, contributes to a better understanding of this multifaceted topic. Of course, there are many other stresses and tolerance mechanisms not addressed in the Special Issue and therefore not addressed in Section 3 of this review.

### 3.1. Gibberellin

Among the genes useful in plant tolerance to abiotic stress, there is the gibberillic acids (GA). Cheng and his colleagues [20] review the GA roles and how manipulating them is possible to improve crops, cereals in particular. Moreover, the review highlights the importance of novel alleles of GA-related genes that do not compromise yield and adaptation, and of using advanced technologies to optimize cereal crops under variable environmental conditions.

GA is, in fact, an essential phytohormone for plant growth and development, affecting height, seed germination, flowering, and stress tolerance. Their biosynthesis involves oxidative enzymes and signalling pathways regulated by “DELLA” proteins [156,157], which act as repressors of GA response. In addition, the GAs interact with hormones such as abscisic acid (ABA), auxin, brassinosteroids (BR), and cytokinin. ABA and GA are antagonists during seed dormancy and germination, while auxin synergistically enhances GA biosynthesis and signalling [20]. These genes have improved agronomic traits such as plant height, abiotic and biotic stress tolerance, and seed germination. For example, semi-dwarfing genes (sd-1 in rice and sdw1/dense in barley) contributed to the “Green Revolution” by increasing lodging resistance and yield. Genes like *GA20ox*, *GA3ox*, and *GA2ox* regulate GA metabolism and are key targets for genetic modification. For instance, mutations in *GA2ox* enhance drought and salinity tolerance by reducing GA levels and promoting stress adaptation. The advanced genome editing technologies, such as the use of CRISPR/Cas9, enable targeted modifications of GA-related genes to improve plant architecture, stress tolerance, and yield, accelerating the breeding process [158,159,160]. Cheng and his colleagues [20] emphasize the need to identify novel GA-related alleles that enhance stress tolerance and yield without adverse effects on other agronomic traits.

### 3.2. Polyamine Seed Priming

Polyamines (PAs) are seed-priming agents that improve plant tolerance to abiotic stresses. Polyamines, such as putrescine, spermidine, and spermine, are bioactive molecules that positively influence germination, seedling growth, and stress resistance. Seed priming with PAs improves photosynthesis, reduces oxidative damage, and increases the activity of antioxidant enzymes. They also modify gene expression related to antioxidant defence, osmole production, and polyamine metabolism. In addition, polyamines interact with DNA, influencing chromatin condensation and epigenetic mechanisms, suggesting a possible role in priming memory. However, the precise molecular mechanisms of PAs priming remain unclear and require further studies. The review by Wojtyla et al. [36] highlights the importance of further molecular analyses and translating research findings into practical applications to improve crop production under stress conditions.

### 3.3. Cadmium Stress

A stress other than strictly due to climate could be due to heavy metals such as cadmium. Heavy metal toxicity in plants manifests through multiple physiological and metabolic disturbances, including growth inhibition, chlorosis, and root necrosis. At the cellular level, heavy metals constrain nutrient and water absorption by altering membrane permeability, perturb the photosynthetic and respiratory electron transport chains in chloroplasts and mitochondria, and disrupt peroxisomal redox homeostasis. These impairments collectively exacerbate the generation of reactive oxygen species (ROS), thereby amplifying oxidative stress and cellular damage [161].

The paper by Li and colleagues [57] is a scientific review that analyses the transcriptional regulatory networks of plants in response to cadmium (Cd) stress, a heavy metal highly toxic for plants and humans. It could be accumulated in agricultural soils and cause severe damage to plant growth, affecting enzymes, photosynthesis, respiration, and membrane systems, leading to tissue necrosis and death. In addition, it represents a risk to human health, entering the food chain and causing diseases such as anaemia, cancer, and kidney failure.

Plants have developed mechanisms to absorb, transport, and detoxify cadmium. Several transporters involved in these processes have been identified, such as OsIRT1, OsNRAMP5, and OsHMA3, which regulate Cd uptake, transport, and chelation. The review also explores the role of transcription factors (TFs) such as WRKY, MYB, bZIP, and HSF, which are involved in several plants’ tolerance and among these modulate gene expression to improve Cd tolerance. Furthermore, it highlights the importance of epigenetic regulation (DNA methylation), lncRNAs, and miRNAs in controlling transcriptional responses to Cd.

The paper also discusses future strategies to reduce Cd accumulation in cereals and improve crop tolerance, such as genetic engineering and CRISPR-Cas9 to develop plant varieties with low Cd accumulation. Finally, it highlights the need for further research to better understand the mechanisms and response of Cd perception; as well as to identify novel genes and TFs in hyper accumulating plants.

### 3.4. Genome-Wide Association (GWA)

Genome-wide association studies represent a powerful approach to dissect the genetic architecture of complex traits, including abiotic stress tolerance, by scanning the entire genome for associations between genetic markers and phenotypic variation in a population [162]. The Special Issue “Molecular Breeding for Abiotic Stress Tolerance in Crops” other than the three above-mentioned reviews, includes six research articles. One of the methodologies often used to study tolerance is the genome-wide association (GWA). The paper by Zhang et al. [65] investigates salt tolerance and yield-related traits in *Brassica napus* through genome-wide association studies (GWAS). Using 10,658 high-quality SNP markers, 77 SNPs associated with salt tolerance and 497 SNPs related to yield were identified. Among them, 19 candidate genes were related to salt tolerance, and seven of them affected both salt tolerance and yield. Salt tolerance was studied at the germination stage, showing significant variations in the traits of germination vigour, germination rate, and salt-related damage index. In addition, six yield-related traits were analysed: silique length, number of siliques on the main inflorescence, seeds per silique, thousand-seed weight, yield per plant, and dry weight per plant. The analysis revealed a complex association between salt tolerance and yield traits, with some pleiotropic loci influencing multiple traits. Candidate genes identified include transcription factors and proteins involved in salt stress regulation and plant growth. These results provide useful genetic resources for breeding programs aimed at salt tolerance.

Ma et al. [91] utilized the GWA to analyse the Membrane Attack Complex and Perforin family (*MACPF*) genes in four species of the *Solanaceae* family (i.e., *Capsicum annuum*, *Solanum lycopersicum*, *Solanum tuberosum*, and *Nymphaea colorata*). MACPFs are proteins involved in plant development and response to environmental stress, but their function in plants is still poorly understood. Twenty-six *MACPF* genes were identified and classified into three main groups based on phylogenetic analyses, gene structures, and domain organization. The expansion of *MACPF* genes in *Solanaceae* shows a non-uniform distribution in the chromosomes without segmental or tandem duplication events. Gene expansion is attributed mainly to dispersed duplications. The genes have undergone purifying selection during evolution. One hundred and ninety-six regulatory elements (CREs) were identified in the promoters of *MACPF* genes, suggesting their involvement in the biotic and abiotic stress responses, such as cold, drought, salinity, and submergence. The *MACPF* genes show preferential expression in reproductive tissues and are regulated by environmental stimuli and phytohormones. In particular, the *CaMACPF1/2/3/4/6* genes were induced by hypoxic stress (submergence), and the *CaMACPF6* gene was localized in the nucleus and plasma membrane, suggesting a specific role in the stress responses. These results represent a basis for future functional investigations of *MACPF* genes and to improve the resistance of crops of the *Solanaceae* family to environmental stresses, contributing to agricultural sustainability.

Another research using GWA is that of Gu et al. [8], which focused on the identification, characterization, and expression analysis of the strictosidine synthase-like (SSL) gene family in maize (*Zea mays* L.). The SSL family is known for its role in synthesizing monoterpene alkaloids, which are important for plant responses to environmental stresses. Twenty genes were identified investigating their distribution, evolution, structure, and function. The identified SSL genes were unevenly distributed across six chromosomes of the maize genome. Tandem duplication was found to be a major factor in the evolution of the SSL gene family in maize, while the phylogenetic analysis grouped 105 SSL genes from maize, sorghum, rice, *Aegilops tauschii*, and *Arabidopsis* into five evolutionary groups. As expected, maize SSL genes showed closer evolutionary relationships with sorghum than with other species. Within the same phylogenetic group exhibited similar exon-intron structures and conserved protein motifs, indicating functional and evolutionary consistency. In addition, promoter regions of the *ZmSSL* genes contained numerous cis-elements related to development, hormone responses, and abiotic/biotic stress, suggesting their involvement in both abiotic and biotic stress adaptation. The genes’ responses to stress were not consistent among plant tissues, nor for all genes. This research contributes to understanding the molecular mechanisms underlying maize stress resilience and offers potential targets for improving crop tolerance to environmental challenges.

### 3.5. Transcription Factors and Gene Silencing

Transcription factors are proteins that regulate gene expression by binding to specific DNA sequences in the promoter or other regulatory regions of a gene. Their main function is to modulate the transcription process, i.e., the synthesis of RNA from DNA, influencing the activation or repression of specific genes. This process helps plants and organisms respond to adverse conditions, such as drought, salinity, or extreme temperatures. Transcription factors could be classified into families based on the structure of their DNA binding domain. Some examples include: (i) MYB, which regulates growth, development, and stress response; (ii) bHLH involved in processes such as cell cycle regulation and stress response; (iii) WRKY has an important role in the biotic and abiotic stress response; and (iv) NAC is involved in regulating development and stress tolerance.

Liu et al. [7] studied the function of the SlMYB78-like transcription factor in tomato and its role in drought and salinity stress tolerance through the ABA pathway. The abscisic acid (ABA) family is highly involved in several plants’ activities and in the tolerance to drought and salt stresses. SlMYB78-like is classified as an R2R3-MYB transcription factor and is highly expressed in flowers, senescent leaves, sepals, and roots. Its expression is induced by drought, salt stress, and exogenous ABA treatments.

MYB transcription factors are one of the largest transcription factor families in plants, with crucial roles in growth, development, and response to abiotic stresses. These factors are classified into four main types, with the R2R3-MYB group playing a particularly complex and significant role. Examples of MYBs include AtMYB2, which regulates drought response genes, and AtMYB49, which increases salt tolerance.

The authors used RNA interference (RNAi) to silence SlMYB78-like and examine its effects on tomato plants under abiotic stress conditions. RNAi lines showed reduced tolerance to drought and salt stress, with lower ABA accumulation, relative water content, and chlorophyll levels, but higher malondialdehyde (MDA) and hydrogen peroxide (H_2_O_2_) levels, indicating increased oxidative stress. Stress-related genes involved in ABA biosynthesis and response were down-regulated, while ABA degradation genes were up-regulated in RNAi lines. Silenced plants exhibited inhibited root and hypocotyl growth, and photosynthetic efficiency was reduced due to the chlorophyll synthesis and photosynthesis-related genes that were significantly down-regulated, under stress treatments.

SlMYB78-like directly binds to the promoter of SlCYP707-A2, an ABA degradation gene, and inhibits its transcription, thereby regulating ABA levels. Protein–protein interaction assays revealed that SlMYB78-like physically interacts with SlDREB3, a dehydration response factor involved in ABA signalling.

The study contributes to understanding plant stress responses and offers potential strategies for breeding tomatoes with enhanced tolerance to abiotic stresses. It provides insights into the molecular mechanisms of stress tolerance and offers a foundation for developing stress-resistant tomato varieties.

### 3.6. Salinity Tolerance

Salt stress introduces into the plant two ions, sodium (Na^+^) and chloride (Cl^−^), which accumulate inside cells; both could have hazardous effects, but Na^+^ accumulation is typically much more dangerous for organ and cell integrity than Cl^−^ accumulation. Consequently, most of the research articles when treating salinity problems deal with Sodium rather than Chloride. In addition, Na^+^ is the predominantly toxic ion for grasses [163], while legume species are particularly sensitive to high levels of Cl^−^, which causes leaf chlorosis [164].

Slabu et al. [165] to clarify whether Na^+^ or Cl^−^ is primarily toxic to faba bean, applied salinity in 25 mM daily increments until the final concentrations of 100 mM NaCl, 100 mM KCl, or 75 mM Na_2_SO_4_ were attained. They concluded that Na^+^ is the principal toxic ion in faba bean. Its accumulation disrupts K^+^ homeostasis, impairs stomatal regulation, and leads to excessive transpirational water loss and tissue necrosis. Concurrently, Cl^−^ contributes to chlorotic toxicity by promoting chlorophyll degradation; however, this effect appears to be contingent upon elevated Na^+^ levels and can be alleviated through supplementary Mg^2+^ application.

Wang et al. [31] describes the salinity tolerance mechanisms in two highly salt-tolerant wheat germplasm lines developed by crosses between moderately tolerant lines, i.e., Chinese Spring (CS)/*Thinopyrum junceum* disomic addition line AJDAj5 (AJ) and the Ph-inhibitor line (Ph-I) derived from CS/*Aegilops speltoides*. AJ shows low sodium levels in leaves, indicative of a sodium exclusion mechanism, while Ph-I and two progeny lines (W4909 and W4910), identified as potential for future breeding programs, show high sodium levels in leaves, indicative of tissue tolerance.

The research focused on two main tolerance mechanisms: sodium exclusion and tissue tolerance. The importance of combining sodium exclusion mechanisms and tissue tolerance to improve salinity tolerance in wheat was highlighted, as well as the need for further research to identify and map the genes responsible for these mechanisms. In addition, four molecular markers (STS) were developed to identify the presence of chromatin of *Aegilops speltoides* chromosome 3B and two progeny lines (7762 and 7159) were identified as the best materials for future research, as they are homozygous for STSs and do not contain the psr1205 marker associated with chromosomal instability.

### 3.7. Gene Editing

Sheng et al. [77] working in rice, which is sensitive to salt stress, detected a novel salinity-tolerant third-generation hybrid rice using CRISPR/Cas9 gene editing to enhance rice productivity in saline-alkali environments by targeting the OsRR22 gene, which negatively regulates salinity tolerance. The edited rice lines exhibit significantly improved salinity tolerance without compromising other agronomic traits. Mutant lines showed better growth under salt stress, with improved fresh weight and seedling length. The fresh weight of the best-performing lines increased by 2.68 and 3.87 times compared to wild-type. The edit efficiency ranged from 59% to 84% depending on the starting background. Two edited lines (i.e., 733Srr22-T1447-1 and HZrr22-T1349-3) with 0 bp and 1 bp deletions, respectively, were developed. Then, by crossing the edited line, a new hybrid rice line was created. It demonstrated promising results in terms of salinity tolerance and yield under salt-stressed conditions, exhibiting a grain yield of 16.22 g per plant in salt-treated plots, a 45% increase compared to control plants. Stress conditions were obtained by using a 0.8% NaCl solution, while field trials involved irrigation with seawater and fresh water. The study highlights the potential of combining CRISPR/Cas9 technology with hybrid rice breeding for improved crop resilience, underlining the absence of transgenes in the T1 generation, which was confirmed through specific PCR tests. So, plants were considered transgene-free since neither Cas9 nor HPT genes were detected.

## 4. Future Perspectives and Concluding Remarks

A comprehensive understanding of the intricate molecular networks governing plant responses to abiotic stresses is paramount for the rational design and implementation of molecular breeding strategies aimed at enhancing crop resilience and productivity in challenging environments [166]. Among the future perspectives, the highest potential is taken by the new genetic engineering, gene and genome editing [57]. CRISPR/Cas9 could be used with several genes involved in stress tolerance [167]. It is already used in several gene editing in the Special Issue was considered the technology used to enable targeted modifications to GA-related genes to improve plant architecture, stress tolerance and yield [20]; to enhance rice productivity in saline-alkali environments [77]; and to develop plants with low cadmium accumulation.

The development of molecular markers to identify the genes and use them for marker assistance selection is another point of attention in developed tolerant plants [31]. In addition, it should be continued to study the gene interactions, the signal regulations [4], and the abiotic signal response to stresses [4]. The understanding of epigenetic regulation is important in several transcriptional responses [36,57] or the identification of the target genes that are subjected to regulation by epigenetic mechanisms and regulatory RNAs and establishing the processes and pathways by which these target-genes influence plant responses is another important issue. Epigenetic was studied to explore the involvement of Polyamines in priming memory through epigenetic mechanisms [36] and the interaction with DNA to facilitate transcription and regulate gene expression [168].

Gene editing is also useful for gene or transcription factors function and/or mechanisms validations, and to better understand their role in response to stresses [5,20,77,91]. These future perspectives emphasize the importance of genetic engineering, stress tolerance, epigenetic studies, functional validation, and practical agricultural applications to address challenges in crop production and environmental stress.

Finally, the extensive amount of data available could be more easily managed and ordered by recent AI technologies and machine learning instruments. Machine learning algorithms excel at handling high-dimensional data, capturing non-linear relationships, and making predictions based on complex interactions, making them particularly well-suited for unraveling the intricacies of plant stress biology. It offers a powerful toolkit for predicting plant stress responses and optimizing agricultural management practices [169]. The conventional plant disease and pest detection methods are time-consuming and labor-intensive. Being insufficient to meet modern agricultural development needs, while deep learning technologies have emerged as a promising solution [170]. The ability to analyze vast datasets from diverse sources, including sensors and imaging equipment, allows for early detection and prediction of various plant health conditions [118]. Furthermore, machine learning models can be trained to predict crop yields based on historical weather patterns, soil conditions, and management practices, providing farmers with valuable insights for optimizing resource allocation and mitigating potential losses [171,172]. Content-based filtering and deep learning techniques are paving the way for more effective and sustainable agricultural practices.

The GRAiN (Gene Regulation and Association Network), an interactive query-based platform, is an example of platform that could facilitate the study of functional relationships between transcription factors (TFs) and genetic modules underlying abiotic-stress responses. It is available at http://rrn.uark.edu/shiny/apps/GRAiN/ (accessed on 15 September 2025) [162].

### 4.1. Breeding Approaches to Improve Plant Tolerance to Abiotic Stresses

To improve plant tolerance to abiotic stress, breeding approaches should use future perspectives. Genomic-assisted breeding (GAB) tools can be utilized, including genome sequencing, quantitative trait loci (QTL) mapping, genome-wide association studies (GWAS), genomic selection (GS), haplotype-based breeding (HBB), epigenomics, and single-cell genomics. These tools enable the identification and manipulation of key genes associated with tolerance, thereby accelerating the development of resistant crop varieties. Furthermore, it is important to integrate high-throughput phenotyping (HTP) and artificial intelligence (AI) technologies to improve the efficiency and accuracy of breeding programs. AI can analyze large amounts of genomic and phenotypic data, predict long-term outcomes, and optimize breeding cycles to enhance plant health and productivity. The analysis of large datasets using machine learning has become indispensable for deciphering the genetic architecture of plants, thereby enabling targeted manipulation of traits critical for crop improvement [173]. This includes leveraging deep learning models, such as convolutional neural networks, for the early and accurate detection of plant diseases and pests, which significantly reduces crop losses and enhances overall agricultural productivity [174,175]. Moreover, AI and machine learning techniques contribute significantly to optimizing resource utilization, such as water and fertilizers, by providing predictive models that guide precision agriculture practices [176,177]. This data-driven approach, incorporating big data analytics, artificial intelligence, and machine learning, fundamentally transforms traditional plant breeding by enabling more precise and efficient selection processes, thereby reducing the time required for trait development [178]. Beyond accelerating trait development, AI also facilitates real-time crop monitoring and management, providing farmers with actionable insights for optimizing crop health and yield [179].

Another crucial aspect is the use of crop wild relatives, which are naturally tolerant to various stresses, to transfer stress-tolerant traits into modern crops. These approaches can be combined with gene editing and transgenomic techniques to improve crop stress tolerance. Moreover, the integration of advanced technologies would accelerate the development of stress-tolerant crops, ensuring agricultural sustainability and global food security [180]. Table 4 summarizes the classical and modern approaches to developing stress-tolerant plants.

### 4.2. Concluding Remarks

Advances in CRISPR-Cas genome editing, RNA interference, omics technologies, nanotechnology, and artificial intelligence have greatly expanded our understanding of plant stress physiology and accelerated the development of resilient crop varieties (Figure 3). Nevertheless, the integration of immune signaling under the combined influence of abiotic and biotic stresses remains only partially understood, particularly at the tissue-specific and cellular scales where critical regulatory events occur. Moreover, significant practical barriers still hinder the translation of these insights into the field, including challenges in delivery systems, regulatory constraints, and the need for long-term validation under diverse environmental conditions.

As climate change intensifies, unraveling the intricate crosstalk between abiotic and biotic stress signaling pathways becomes increasingly vital for achieving sustainable agriculture and safeguarding global food security. Looking ahead, future directions emphasize the development of real-time monitoring and high-resolution analytical approaches—such as single-cell omics and spatial transcriptomics—which hold promise for fine-tuning plant immune responses and enabling precision crop improvement strategies.

Furthermore, the adoption of advanced AI tools, including big data analytics and the Internet of Things, offers critical solutions for enhancing agricultural productivity, sustainability, and resource efficiency [181]. These technologies allow for the automation of processes and real-time monitoring, leading to more informed decision-making in agricultural management [182].

Modern breeding technologies such as CRISPR-Cas9, marker-assisted selection, and advanced genomic analyses have revolutionized plant improvement by allowing precise and targeted modification of genes. These tools dramatically accelerate the breeding process, reducing the time required to introduce desirable traits such as drought tolerance, salinity resistance, or enhanced disease resistance from several years to just a few months. By enabling scientists to directly edit or track genes associated with stress resilience and yield stability, these approaches not only improve the efficiency of breeding programs but also contribute substantially to strengthening global food security.

However, despite these remarkable advances, traditional breeding methods remain indispensable. They provide a broad genetic base and preserve natural diversity, which is essential for the long-term adaptability of crops to changing environments and emerging challenges such as climate variability, novel pathogens, and shifting soil conditions. While molecular tools can fine-tune specific traits with precision, traditional approaches capture the complexity of polygenic traits and maintain the ecological balance within plant populations.

The future of sustainable agriculture will therefore depend on the integration of molecular innovations with conventional breeding wisdom. By combining the speed and accuracy of cutting-edge technologies with the resilience and diversity offered by traditional practices, agriculture can evolve into a system that is both highly productive and ecologically sustainable. Such an integrated approach ensures that while we harness scientific innovation for rapid improvement, we also safeguard the genetic diversity and adaptability that plants have developed over millennia. This synergy will be essential to secure food supplies for a growing global population, especially under the uncertainties posed by climate change and resource limitations.

## Figures and Tables

**Figure 1 ijms-26-09164-f001:**
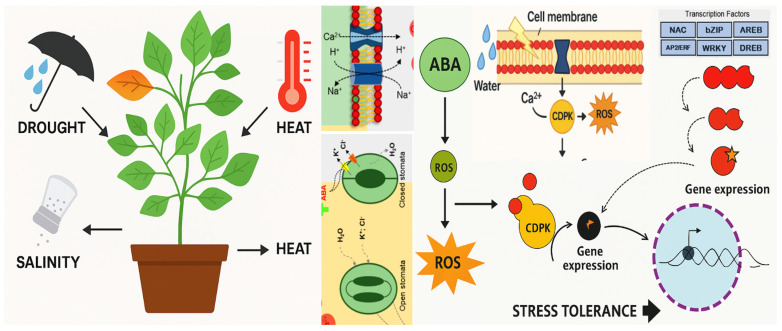
Abiotic stresses and different plant reactions to tolerate the stresses.

**Figure 2 ijms-26-09164-f002:**
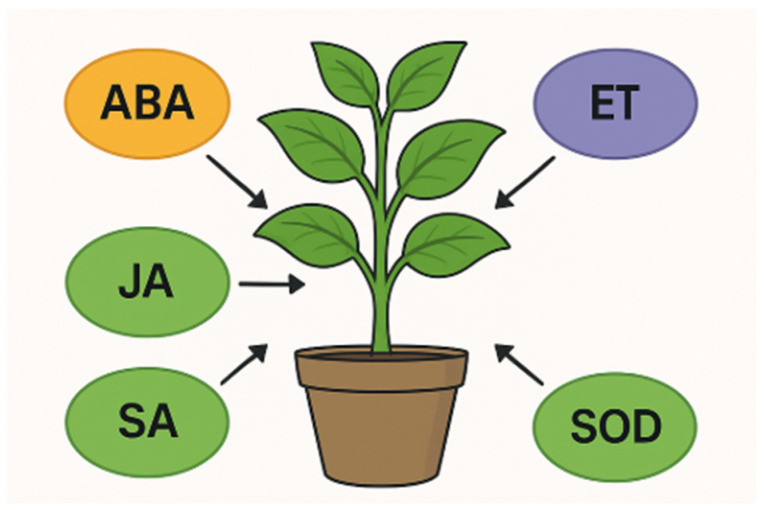
Hormonal regulation involved in stress responses.

**Figure 3 ijms-26-09164-f003:**
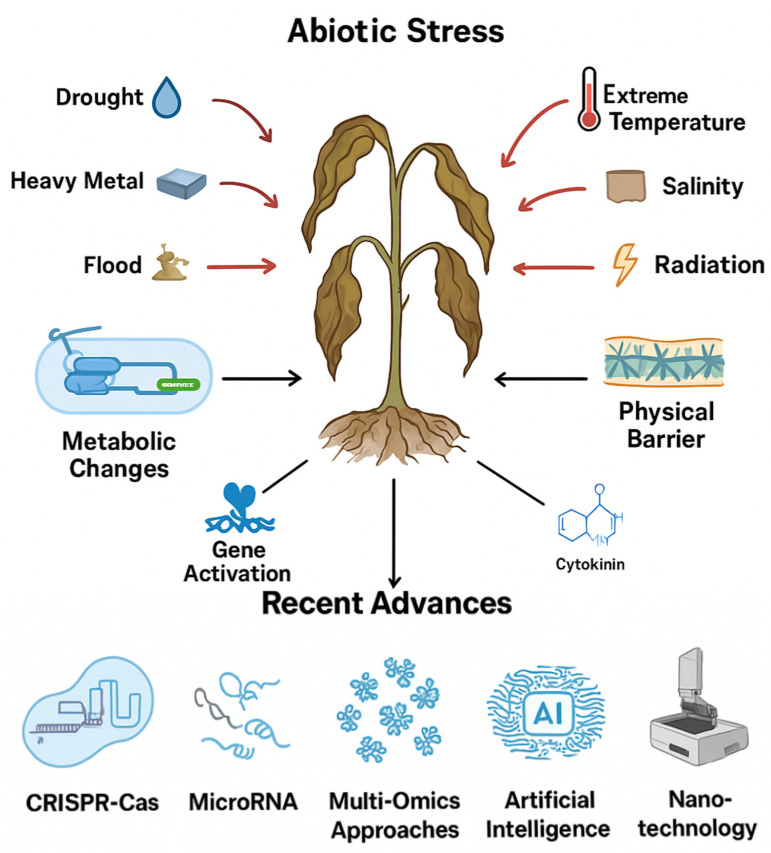
Stress (in red) pathway and list of recent tools to manage plant tolerance.

**Table 1 ijms-26-09164-t001:** Some examples to improve tolerance using CRISPR/Cas9.

Pianta	Gene	Stress	Effect of Editing	Reference
Rice	*OsRR22*	Salinity	Mutants with improved salt tolerance, without compromising performance	[76,77]
	*OsDST*	Drought and Salt	Greater tolerance to drought and salinity	[77,78]
	*OsDERF1*	Drought	enhanced drought resistance	[79]
	*OsERA1*	Drought (via ABA)	Mutants with longer roots and better response to drought	[80]
	*OsPYL1/4/6*	Hoy	Knock-out improves heat tolerance and productivity	[4,81]
Wheat	*TaERF3*	Drought	Increased water efficiency and photosynthesis under stress	[4,82]
	*TaHKT1;5*	Salt	Improved ionic regulation and salt tolerance	[20,31,36,57,83]
	*TaDREB2/3*, *TaHAG1*	Salt/drought	Promotes tolerance, improved multi-source resistance	[4,5,57,84]
Barley	*HvITPK1*	Salt	Inositol Trisphosphate 5/6 Kinase	[85]
Maize	*ZmARGOS8*	Drought	Improve grain yield	[86]
	*ZmTMS5*	Thermosensitive		[87]
Soybean	*GmNAC8*, *GmNAC12*	Drought	Mutanti più sensibili = ruolo positivo nella tolleranza	[5,31,57,77,88]
	*GmMYB118*, *GmAITR*	Salt/drought	Overexpression → positive role in tolerance	[89]
Tomato	*SlARF4*, *SlHyPRP1*	Salt/drought	Knock-out improves water efficiency and stress tolerance	[90,91]
	*SICBF1*	Cold	C-repeat binding factors (CBFs)	[92]
	*SlMAPK3*	Hot and Salt	Heat-resistant mutants, with increased HSP; salinity sensitive	[93]
Apple	*MdNHX1*	Salt	Salt tolerance high K^+^/Na^+^ in leaves	[94]
Cotton	*GhHB12*	Salt/drought	Increases the tolerance to abiotic stress	[95]
Mustard	*BnaA6.RGA*	Drought	Interacting with the ABA signaling	[96]
Chickpea	*4CL*	Drought	phenylpropanoid metabolism in the lignin biosynthesis pathway	[97]
	*REV7*	Drought	MYB transcription factor	[97]
Actinidia eriantha	*AcePosF21*	Cold	bZIP transcription factor	[98]
	*AceGGP3*	Drought	MYBS1-like and GBF3 transcription factors. AsA	[99]
Peanut	*AVP1*	Drought/salt	Biomass, photosynthetic rate, high yield	[100]
Cassava, cucumber, Tomato	*eIF4E, CsLOB1, SlDmr6-1*	Virus or bacteria	Broad resistance through gene knockouts	[101,102,103]
Poplar	*CarNac3*	Drought	Transcription factor from chickpea	[104,105]
	*PagHCF106*	Drought	Modulating stomatal aperture	[106]
	*PagHB7/PagABF4–PagEPFL9*	Drought	Regulates Stomatal Density	[107]

**Table 2 ijms-26-09164-t002:** Some examples of biostimulants applications.

Biostimulant	Application	Main Effects	Crops	Ref.
Seaweed extracts (Ascophyllum nodosum, Ecklonia maxima)	Foliar spray, soil drench	Promote root and shoot growth, enhance photosynthesis, improve tolerance to drought, salinity, and temperature stress	Wheat, tomato, maize	[135]
Humic and fulvic acids	Soil amendment, fertigation	Improve nutrient uptake (N, P, Fe, Zn), stimulate root development, enhance soil microbial activity	Maize, soybean, cucumber	[136]
Protein hydrolysates and amino acids	Foliar spray, seed treatment, fertigation	Stimulate N metabolism, increase photosynthetic efficiency, improve yield and fruit quality under stress	Tomato, lettuce, grapevine	[137]
Microbial biostimulants (PGPR: *Azospirillum*, *Bacillus*, *Pseudomonas*; *Mycorrhiza*: *Glomus* spp.)	Seed coating, soil inoculation, root dipping	Enhance nutrient uptake (P, N), improve root architecture, boost stress tolerance and pathogen resistance	Maize, wheat, legumes	[138]
Chitosan and biopolymers	Foliar spray, seed coating	Act as defense elicitors, strengthen antioxidant response, improve resistance to pathogens	Strawberry, tomato, pepper	[139]
Silicon-based compounds	Foliar spray, soil application	Reinforce cell walls, reduce lodging, increase tolerance to salinity, drought, and heavy metals	Rice, cucumber, barley	[140]

**Table 4 ijms-26-09164-t004:** Models representing an innovative and integrated approaches to develop stress-resistant crops, combining advanced technologies and traditional breeding strategies to ensure agricultural sustainability and global food security.

Typology	Method
Integrating genomics and advanced phenotyping	Breeding-assisted genomics (GAB). Using tools such as genomic selection (GS), quantitative trait loci (QTL) mapping, genome-wide association studies (GWAS), and haplotype-based breeding (HBB) to identify and select the best genotypes.
	High-throughput phenotyping (HTP): Non-destructive technologies to assess plant phenotypic characteristics, such as stress tolerance, quickly and accurately.
Using Artificial Intelligence (AI)	Predictive models: AI is used to analyze genetic and phenotypic data, predict breeding outcomes, and identify the most promising lines.
	Automation: AI accelerates the selection process and optimizes backcrossing cycles to combine desirable traits such as stress tolerance and high yield.
Pan-genomics and super pan-genomics	Pangenomic analysis: Study of genetic diversity within a species to identify key genes associated with stress tolerance.
	Super pan-genomics: Using high-resolution genomic data to discover rare and structural variants that can be integrated into breeding programs.
De novo domestication and use of wild relatives	Wild relatives and halophytes: Harnessing the genetic diversity of naturally stress-tolerant plants to transfer useful traits into modern crops.
	De novo domestication: Genetically modifying wild plants to make them cultivable and resistant to stress.
Accelerated breeding pipeline	Speed breeding: Using controlled environments to accelerate plant growth and reproduction cycles.
	AI-assisted backcrossing: Predictive models to select the best lines during backcrossing cycles.
“Horses for courses” approach	Combination of specific genes: Customization of genetic combinations based on specific stress levels and environmental conditions.
Multi-location and station tests	Large-scale evaluation: Testing of advanced lines under different environmental conditions to ensure the stability and adaptability of the traits.

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
