# Peer review of "Molecular Breeding for Abiotic Stress Tolerance in Crops: Recent Developments and Future Prospectives"

_ijms, 2025, doi:10.3390/ijms26189164_

Round 1
Reviewer 1 Report (Previous Reviewer 1)
Comments and Suggestions for Authors|
4 |
87 |
1.2. Antioxidant Defence |
Need updating |
|
4 |
107 |
1.3. Osmotic Adjustment |
Need updating data. Not only proline and GB but also there several compounds do the same role (ie. T6P) |
|
3 |
95 |
sodium accumulation |
Which is more hazard for organs in cell Na or Cl accumulation? |
|
7 |
223 |
Cadmium Stress |
Where other heavy metal stress such AL, Lead, arsenic..etc? |
|
7 |
240 |
The paper also discusses future perspectives |
What are these perspectives? |
|
8 |
265 |
Ma et al. [55] utilize the GWA…. |
Author should mention the name? |
|
10 |
357 |
3.7. Gene editing |
Author discussed effect of salinity (by using Nacl) based on fact that Na only element that poison for plant meanwhile impact of up take Cl is more hazard than Na |
Author Response
Rev 1
I would like to thank Rev1 for their review work. I report hereafter the Rev1 comments each one followed by my answer/specification.
|
4 |
87 |
1.2. Antioxidant Defence |
Need updating |
Thanks for the valuable suggestion. I updated and enlarged, especially the non-enzymatic antioxidant’s role, inserting a most recent references of 2025.
|
4 |
107 |
1.3. Osmotic Adjustment |
Need updating data. Not only proline and GB but also there several compounds do the same role (ie. T6P) |
Thanks for the valuable suggestion. I updated and enlarged, especially giving more details on osmolytes and their role, inserting more recent references of 2024 and 2025. I also enlarge the list of osmolytes (proline, glycine betaine (GB), amino acids, sugars (e.g. Trehalose, T6P), alcohols, and quaternary ammonium).
|
3 |
95 |
sodium accumulation |
Which is more hazard for organs in cell Na or Cl accumulation? |
Thanks for the appropriated question. Both sodium (Na⁺) and chloride (Cl⁻) accumulation inside cells can be hazardous, but Na⁺ accumulation is typically much more hazardous for organ and cell integrity than Cl⁻ accumulation. For this most of the papers when treating salinity problems deal with Sodium rather than Chlorite. I inserted this concept in section 3.6 enlarging a bit the toxicity effects due to the two ions.
|
7 |
223 |
Cadmium Stress |
Where other heavy metal stress such AL, Lead, arsenic..etc? |
Thanks for the appropriate question/comment. Of course, yes there are several other stresses. I mentioned only cadmium since the section “3. Molecular Breeding for Abiotic Stress Tolerance in Crops” deals only with papers published within that special issue. To make it clearer I add at the beginning of section 3 the following text: “Of course, there are many other stresses and tolerance mechanisms not addressed in the special issue and therefore not addressed in section 3 of this review.” In addition to fill your valuable point, which could be the same of other reader, I inserted also a general sentence on heavy metal toxicity in plants as a whole to consider all the heavy metals.
|
7 |
240 |
The paper also discusses future perspectives |
What are these perspectives? |
The future prospectives are the use of genetic engineering and CRISPR-Cas9 as mentioned at the end of the sentence. I changed “proposing” with “such as” and moved “strategies” in the place of “prospectives” to make the sentence clearer. I wish this solve the problem. I also added nanotechnologies among the possible prospectives, and enlarged the section.
|
8 |
265 |
Ma et al. [55] utilize the GWA…. |
Author should mention the name? |
I am sorry I don’t understand the question and which is the “name” I should mention. Citation is with surname no name is necessary, so it seems to agree with the Journal Style. While GWA acronyms (Genome-wide association) is already disclosed in the section 3.4 name.
|
10 |
357 |
3.7. Gene editing |
Author discussed effect of salinity (by using Nacl) based on fact that Na only element that poison for plant meanwhile impact of up take Cl is more hazard than Na |
I partially answer to this in your third point and inserted that in section 3.6 when treating salinity (please see previous answer). Hereafter the reference
“For most species, Na+ appears to reach a toxic level before Cl− does, and probably for this reason most studies have been focused on Na+ exclusion and the controlling of Na+ transport within the plant [11,12]” from Behzadi Rad, P.; Roozban, M.R.; Karimi, S.; Ghahremani, R.; Vahdati, K. Osmolyte Accumulation and Sodium Compartmentation Has a Key Role in Salinity Tolerance of Pistachios Rootstocks. Agriculture 2021, 11, 708. https://doi.org/10.3390/agriculture11080708].
I would like to thank Rev 1 for their valuable suggestion to improve my manuscript
Reviewer 2 Report (New Reviewer)
Comments and Suggestions for Authors
Reviewer - Comments
I got acquainted to your work entitled “Molecular Breeding for Abiotic Stress Tolerance in Crops: Recent developments and Future prospectives” with great interest.
This study encompasses the review version of the special issue “Molecular Breeding for Abiotic Stress Tolerance in Crops” published in the Int. J. Mol. Sci. It reviews molecular breeding strategies to enhance abiotic stress tolerance in crops, addressing challenges like drought, salinity, temperature extremes, and waterlogging, which threaten global food security.
Overall, the manuscript is interesting, but no innovation was found. However, I believe that it could not be published in its current form. Therefore, there are some points that need to be addressed prior publication. Respecting your hard work, I suggest a “major revision” for this manuscript. I encourage you to seriously consider following concerns.
Major concerns:
- Authors described the data related to the transcription factors, hormones, genome assisted breeding etc. This type of data could be easily accessed and the extensive amount of data already available in the literature.
- Authors should put more focus on the studies containing recent technologies e.g. genome editing and AI/ machine learning. That would be some value-addition to the literature.
- Authors only included gene editing examples of very few crops, there’re not enough examples for fruits, trees e.g. Populus or cash crops e.g. Cotton. I encourage authors to provide a variety of data regarding this heading.
- Likewise, machine learning topic is very short and unclear. Please elaborate on this topic with some practical literature examples.
Author Response
Rev2
I would like to thank Rev2 for their review work. I report hereafter the Rev2 comments each one followed by my answer/specification.
I got acquainted to your work entitled “Molecular Breeding for Abiotic Stress Tolerance in Crops: Recent developments and Future prospectives” with great interest.
This study encompasses the review version of the special issue “Molecular Breeding for Abiotic Stress Tolerance in Crops” published in the Int. J. Mol. Sci. It reviews molecular breeding strategies to enhance abiotic stress tolerance in crops, addressing challenges like drought, salinity, temperature extremes, and waterlogging, which threaten global food security.
Overall, the manuscript is interesting, but no innovation was found. However, I believe that it could not be published in its current form. Therefore, there are some points that need to be addressed prior publication. Respecting your hard work, I suggest a “major revision” for this manuscript. I encourage you to seriously consider following concerns.
Major concerns:
- Authors described the data related to the transcription factors, hormones, genome assisted breeding etc. This type of data could be easily accessed and the extensive amount of data already available in the literature.
The main Review aim is to bring not the data itself, but the general and more important results present in the vast literature in a single place.
- Authors should put more focus on the studies containing recent technologies e.g. genome editing and AI/ machine learning. That would be some value-addition to the literature.
Thanks for your valuable comment. I have made extensive revisions to both sections 2.1 and 2.2, as well as section 4. New concepts were introduced, and some specifications and very updated references were added. ”
- Authors only included gene editing examples of very few crops, there’re not enough examples for fruits, trees e.g. Populus or cash crops e.g. Cotton. I encourage authors to provide a variety of data regarding this heading.
Thanks for your valuable comment, I added references for Barley, Maize, Apple, Peanut, Cotton, Brassica, Chickpea, Kiwi and Poplar.
- Likewise, machine learning topic is very short and unclear. Please elaborate on this topic with some practical literature examples.
Thanks for your valuable comment and suggestion. As mentioned above, I have made extensive revisions to both sections 2.1 and 2.2, as well as section 4. In addition, a description of machine learning, and some examples with updated references were added. ”
I would like to thank you for your consideration and your suggestions to improve the manuscript.
Reviewer 3 Report (New Reviewer)
Comments and Suggestions for Authors
Review of the ms. Molecular Breeding for Abiotic Stress Tolerance in Crops: Recent developments and Future prospectives jms-3843202
The manuscript is comprehensive, relevant, and timely, but several sections are list-like, repetitive, and lacking in examples. I suggest some comments fot the improvement like reduce redundancy, remphasize hormonal cross-talk and integrated networks.
Specific Comments:
Hormonal Regulation
- There is a sisproportion of hormone description. For example ABA is presented in detail, but the other hormones (ET, SA, JA, auxin, cytokinin, GA) are only briefly listed. I suggest to describe these hormones more detail.
- This gives the reader the impression that the balance is lacking.
- I suggest to complement this section with the newest results on hormonal crosstalk of ABA, JA, SA and Et. Highlighting that these phytohormone pathways operate not only individually but also through integrated transcriptional regulatory networks (I suggest to summarize the following publications in this field: https://pmc.ncbi.nlm.nih.gov/articles/PMC11395473/)
- I suggest to write a paragraph on the recent knowledge on the phytohormones in plant development-abiotic stress. In this section you can summarize, that phytohormones are not static, isolated phenomena, but precisely coordinated systems which have function in the context of development and stress (https://www.mdpi.com/1422-0067/26/13/6455).
- 2. It would be useful to clearly describe what it depicts and how it relates to the text.
Gibberellin:
- Too general wording. Many GA reviews tend to simply state that "GA promotes growth, but decreases under stress." This alone is insufficient.
- Please emphasize that GA is often antagonistic to ABA (e.g., in seed germination and drought tolerance regulation) and interacts with auxins in cell elongation and root development.
Furthermore:
- It is strongly recommended to include a dedicated subsection on ecological approaches and nature-based solutions in the manuscript. While the current review provides a comprehensive overview of molecular breeding, hormonal regulation, and genomic tools, it lacks coverage of sustainable, ecological strategies that are increasingly recognized as complementary solutions in enhancing abiotic stress tolerance. These section can include the application of plant-derived biostimulants (e.g. https://www.mdpi.com/2073-4395/15/4/991).
In conclusion, the manuscript addresses an important and timely topic. To further strengthen it, I recommend incorporating recent literature on phytohormone cross-talk and transcriptional regulation, emphasizing GA–ABA antagonism and GA–auxin interactions, and adding a section on ecological and nature-based approaches.
These revisions are important to improve the balance, depth, and impact of the review. Therefore, I recommend major revision.
Author Response
I would like to thank Rev3 for their review work. I report hereafter the Rev3 comments each one followed by my answer/specification.
Review of the ms. Molecular Breeding for Abiotic Stress Tolerance in Crops: Recent developments and Future prospectives jms-3843202
The manuscript is comprehensive, relevant, and timely, but several sections are list-like, repetitive, and lacking in examples. I suggest some comments fot the improvement like reduce redundancy, remphasize hormonal cross-talk and integrated networks.
I would like to thank for your consideration. I revised most of the document, added examples and integrate it following your broad suggestions.
Specific Comments:
Hormonal Regulation
- There is a sisproportion of hormone description. For example ABA is presented in detail, but the other hormones (ET, SA, JA, auxin, cytokinin, GA) are only briefly listed. I suggest to describe these hormones more detail.
- This gives the reader the impression that the balance is lacking.
- I suggest to complement this section with the newest results on hormonal crosstalk of ABA, JA, SA and Et. Highlighting that these phytohormone pathways operate not only individually but also through integrated transcriptional regulatory networks (I suggest to summarize the following publications in this field: https://pmc.ncbi.nlm.nih.gov/articles/PMC11395473/)
- I suggest to write a paragraph on the recent knowledge on the phytohormones in plant development-abiotic stress. In this section you can summarize, that phytohormones are not static, isolated phenomena, but precisely coordinated systems which have function in the context of development and stress (https://www.mdpi.com/1422-0067/26/13/6455).
- 2. It would be useful to clearly describe what it depicts and how it relates to the text.
Thanks for your valuable comment and suggestion
ET, SA, JA, auxin, cytokinin, GA, descriptions are extended. The paragraph with crosstalk has bene improved and a paragraph with your interesting find was added.
Gibberellin:
- Too general wording. Many GA reviews tend to simply state that "GA promotes growth, but decreases under stress." This alone is insufficient.
- Please emphasize that GA is often antagonistic to ABA (e.g., in seed germination and drought tolerance regulation) and interacts with auxins in cell elongation and root development.
As stated in the first paragraph of section 3. This section is devoted to summarizing the papers published in the SI. So, the 3.1. Gibberellin reflects Cheng and his colleagues paper former 9. In any event I added the suggested information.
C Furthermore:
- It is strongly recommended to include a dedicated subsection on ecological approaches and nature-based solutions in the manuscript. While the current review provides a comprehensive overview of molecular breeding, hormonal regulation, and genomic tools, it lacks coverage of sustainable, ecological strategies that are increasingly recognized as complementary solutions in enhancing abiotic stress tolerance. These section can include the application of plant-derived biostimulants (e.g. https://www.mdpi.com/2073-4395/15/4/991).
Thanks very much for the excellent suggestion. The section 2.3 dealing with NBS and biostimulants was added.
In conclusion, the manuscript addresses an important and timely topic. To further strengthen it, I recommend incorporating recent literature on phytohormone cross-talk and transcriptional regulation, emphasizing GA–ABA antagonism and GA–auxin interactions, and adding a section on ecological and nature-based approaches.
These revisions are important to improve the balance, depth, and impact of the review. Therefore, I recommend major revision.
Thanks for your suggestions and indications very useful to improve the manuscript.
Round 2
Reviewer 2 Report (New Reviewer)
Comments and Suggestions for Authors
The authors have answered the questions that were brought up in the last review. The changes, especially those made to the advanced technologies and conclusion sections, have made the manuscript much clearer and more rigorous. The article is now well-organized and adds something useful to the field. I think this paper should be published as it is.
Reviewer 3 Report (New Reviewer)
Comments and Suggestions for Authors
The authors have adequately addressed my previous comments and revised the manuscript accordingly. In its current form, the manuscript is sufficiently improved, and I consider it suitable for publication.
This manuscript is a resubmission of an earlier submission. The following is a list of the peer review reports and author responses from that submission.
Round 1
Reviewer 1 Report
Comments and Suggestions for Authors|
page |
Line |
sentence |
Comment |
|
2 |
72 |
1.2. Antioxidant Defence |
Need updating |
|
2 |
79 |
1.3. Osmotic Adjustment |
Need updating data. Not only proline and GB but also there several compounds do the same role (ie. T6P) |
|
3 |
95 |
sodium accumulation |
Which is more hazard for organs in cell Na or Cl accumulation? |
|
4 |
149 |
Cadmium Stress |
Where other heavy metal stress such AL, Lead, arsenic..etc? |
|
5 |
166 |
The paper also discusses future perspectives |
What are these perspectives? |
|
5 |
188 |
Ma et al. [38] utilize the GWA…. |
Author should mention the name? |
|
6 |
280 |
3.7. Gene editing |
Author discussed effect of salinity (by using Nacl) based on fact that Na only element that poison for plant meanwhile impact of up take Cl is more hazard than Na |